# A 3D-Printed Polymer–Lipid-Hybrid Tablet towards the Development of Bespoke SMEDDS Formulations

**DOI:** 10.3390/pharmaceutics13122107

**Published:** 2021-12-07

**Authors:** Bryce W. Barber, Camille Dumont, Philippe Caisse, George P. Simon, Ben J. Boyd

**Affiliations:** 1Drug Delivery, Disposition and Dynamics, Monash Institute of Pharmaceutical Sciences, Monash University, 381 Royal Pde, Parkville, Melbourne 3052, Australia; bryce.barber@monash.edu; 2Gattefossé SAS, 36 Chemin de Genas, CEDEX, 69804 Saint-Priest, France; cdumont@gattefosse.com (C.D.); pcaisse@gattefosse.com (P.C.); 3Department of Materials Science and Engineering, Monash University, Clayton, Melbourne 3800, Australia; George.Simon@monash.edu; 4Department of Pharmacy, University of Copenhagen, Universitetsparken 2, 2100 Copenhagen, Denmark

**Keywords:** lipid, SMEDDS, additive manufacturing, controlled release

## Abstract

3D printing is a rapidly growing area of interest within pharmaceutical science thanks to its versatility in creating different dose form geometries and drug doses to enable the personalisation of medicines. Research in this area has been dominated by polymer-based materials; however, for poorly water-soluble lipophilic drugs, lipid formulations present advantages in improving bioavailability. This study progresses the area of 3D-printed solid lipid formulations by providing a 3D-printed dissolvable polymer scaffold to compartmentalise solid lipid formulations within a single dosage form. This allows the versatility of different drugs in different lipid formulations, loaded into different compartments to generate wide versatility in drug release, and specific control over release geometry to tune release rates. Application to a range of drug molecules was demonstrated by incorporating the model lipophilic drugs; halofantrine, lumefantrine and clofazimine into the multicompartmental scaffolded tablets. Fenofibrate was used as the model drug in the single compartment scaffolded tablets for comparison with previous studies. The formulation-laden scaffolds were characterised using X-ray CT and dispersion of the formulation was studied using nephelometry, while release of a range of poorly water-soluble drugs into different gastrointestinal media was studied using HPLC. The studies show that dispersion and drug release are predictably dependent on the exposed surface area-to-volume ratio (SA:V) and independent of the drug. At the extremes of SA:V studied here, within 20 min of dissolution time, formulations with an SA:V of 0.8 had dispersed to between 90 and 110%, and completely released the drug, where as an SA:V of 0 yielded 0% dispersion and drug release. Therefore, this study presents opportunities to develop new dose forms with advantages in a polypharmacy context.

## 1. Introduction

Mass manufacturing of oral formulations fails to produce drug forms that adequately address the complexity of an individual’s disease state, age and genetic factors [1,2,3]. This “one-size-fits-all” approach has led to dose-related adverse drug reactions (ADRs), particularly for older patients using polypharmacy [4,5,6]. Recent investigations into the use of three-dimensional printing (3DP) technology as an alternative additive manufacturing method [7,8,9,10,11,12], shows great potential for delivering bespoke oral drug medications. 

3DP encompasses numerous techniques based around the automated and sequential layering of material to construct an object, based on a digitally modelled design. A noteworthy appeal of 3DP in pharmaceutics can be attributed to the potential to generate oral dosage forms in a rapid, reproducible and flexible manner. To date, the realm of 3DP in pharmaceuticals has been dominated by techniques including inkjet/binder jetting [13], selective laser sintering (SLS) [14,15], and fused deposition modelling (FDM) [16,17,18].

FDM is a widely used 3DP technique across multiple disciplines that requires a solid filament to be melted at high temperatures, for extrusion through a small orifice on a layer-by-layer pattern and solidification to produce solid models. In medicine, and more recently in pharmaceutics, this technique has seen use in the production of hydrolysable and biodegradable polymer-based scaffolds [19,20,21]. However, it is a technique that has been largely avoided in the 3DP of pharmaceutics due to the thermolabile nature of most drugs at temperatures required for its extrusion. Recently, FDM has gained greater interest with the development of new polymers that may be coextruded, with a drug, at lower temperatures to avoid thermal degradation during the extrusion process [21,22].

The oral bioavailability of lipophilic drugs can be improved with the coadministration of lipids in the form of food- or lipid-based formulations [23,24]. Personalising lipid-based formulations would represent an important advance in treatments with lipophilic small molecule drugs. However, despite the physicochemical properties of many lipids allowing their extrusion at low temperatures, there is a lack of research into the extrusion of lipid-based formulations in the context of 3DP [25]. Even less is understood around the interplay between surface area-to-volume ratio, dispersion and digestion of such formulations which will dictate release of the drug and rate of absorption for permeable lipophilic compounds.

Lipid extrusion is already recognised as a pharmaceutical manufacturing technique, and creamy spheroids have been produced from the extrusion and pelletisation of a lipid-based solid self-emulsifying drug delivery system (S-SEDDS) [26]. In 2019, Vithani et al. demonstrated a novel approach to the 3DP of thermolabile and lipophilic drugs loaded into solid lipid-based formulations [27]. Vithani employed semi-solid extrusion (SSE) to 3D print solid oral tablets from a lipid-based solid self-micro emulsifying drug delivery system (S-SMEDDS) as the extrudate. In addition, Chatzitaki et al. also employed similar methods to produce a solid lipid-based suppository dosage form via SSE [28]. These methods did not require high temperatures or solvent evaporation; however, the final dosage forms had poor printing fidelity which presents concern for the consistency of behaviour upon ingestion.

It has also been more than a decade since research investigating ‘polypill’ formulations has been conducted to combat ADR and adherence issues associated with polypharmacy [29,30]. Recent research using primarily polymer-based formulations suggest that it is possible to 3DP oral dosage forms with multiple drugs and different release profiles [15,31,32,33,34,35,36]. Furthermore, Markl et al. demonstrated that a 3D-printed dissolvable polymer scaffold could be exploited to generate dual compartmented tablets containing a liquid lipid formulation [36]. However, there is a lack of research in the viability of lipid-based 3DP ‘polypills’ with tuneable drug release especially utilising solid lipid systems.

Therefore, in this study, we have evaluated the control over formulation dispersion and drug release using 3D-printed single- and multi-compartment polymer scaffolds filled with dispersible lipid formulations. The ‘polymer lipid hybrid’ (PLH) tablets were evaluated for the effect of exposed surface area; dissolvable vs non-dissolvable printed scaffolds; as well as multicompartment systems for asynchronous release, on the dispersion of the lipid formulation filled into the compartments of the scaffold (Figure 1). Firstly, single-compartment polymer scaffolds with systematically varying exposed surface area were 3D-printed and filled with an S-SMEDDS formulation loaded with fenofibrate (FEN) as the model poorly soluble drug. The polymer scaffolds were prepared from either poly-lactic acid (PLA, not degradable in the gut on a physiological timescale) and polyvinyl alcohol (PVOH, dissolves on physiological timescales). Subsequently, as a proof of concept, multi-compartment scaffold systems were 3D-printed with six ‘wedges’ forming the circular scaffold (Figure 1), and the wedges were filled individually with the six different combinations of three drugs (clofazimine, lumefantrine and halofantrine) in two different S-SMEDDS formulations. These studies serve as a proof of concept for a PLH system that may enable better control over the fidelity of the lipid formulation in a 3D-printable format and provide great versatility in release of different drugs to address the problems with polypharmacy.

Dispersion studies of the formulations from PLA and PVOH scaffolds were conducted within simulated gastric to intestinal media. Nephelometry and HPLC analytical techniques were employed to determine the rate of dispersion and drug release, respectively. 

## 2. Materials and Methods

Poly-lactic acid filament (Ultimaker PLA, sourced from Imaginables, Melbourne, Australia), poly-vinyl alcohol filament (Ultimaker PVA, sourced from Imaginables, Melbourne), FaSSIF powder (FaSSIF/FeSSIF/FaSSGF, 58 g, 25 L size, Biorelevant.com Ltd, London, United Kingdom), clofazimine and lumefantrine (Sigma, Burlington, VT, USA), halofantrine HCl (SmithKline Beecham Pharmaceuticals, Mysore, India), Gelucire® 44/14 and Gelucire® 48/16 (Gattefossé, Saint-Priest, France), Kolliphor P 188 and NaOH (Sigma Aldrich, Burlington, USA), M4670 silicone rubber mixture (Barnes, Melbourne, Australia), sodium dihydrogen orthophosphate (NaH_2_PO_4_, AnalaR—Merck, Kenilworth, NJ, USA), NaOH pellets and acetonitrile gradient grade (Ajax Finechem (Univar), Scoresby, Australia).

### 2.1. 3D Printing PLA and PVOH Scaffolds

A fused deposition modelling (FDM)-based printer was used to generate physical 3D scaffolds from commercial poly-lactic acid (PLA) and poly-vinyl-alcohol (PVOH) filaments. A computer aided drawing (CAD) software tool ‘Sketchup’ was used to virtually model and export scaffolds as stereolithography files (’.stl’ extension). The stereolithography files were then processed (sliced) with Ultimaker Cura software (v4.6.1–‘Master’, 2020, Ultimaker, Utrecht, Netherlands) using custom slicing profiles. The printing of the PLA and PVOH scaffolds was with a layer height of 0.1 mm, and a print speed of 40 mm/s. These processed models were saved as a G-Code file onto a secure digital (SD) card (‘.gcode’ extension). An Ultimaker 2 3D printer (U2) equipped with a 0.6 mm diameter nozzle was prepared for printing with the desired filament. The U2 build plate was set to 60 °C for 5 min before extruding a small amount of filament from the nozzle at ~215 °C to ensure smooth and featureless filament extrusion. The G-Code file was then manually selected from the inserted SD card in an Ultimaker 2 3D printer and once printing had begun the printing speed was manually set to 80%.

Table 1, Table 2 and Table 3 below present the geometric properties for the 3D-printed scaffolds used in this study. The images show the virtual models designed for PLA-based scaffolds, with properties of the PVOH-based scaffolds underneath. Each of the multicompartment scaffolds (MCTs) and scaffold containers A, B and C (when covered) possess the same target internal volume of ~393 µL. The printing times have been calculated by multiplying the Ultimaker Cura printing time prediction by the print speed dilation (1.25 at 80%).

### 2.2. Preparing Lipid Filled 3DP Scaffolds

#### 2.2.1. Preparation of Drug-Loaded SMEDDS

The polymer–lipid hybrid tablets contained a drug-loaded SMEDDS formulation, which was manually injected as molten liquid into the 3DP scaffold. The dispersion properties of the SMEDDS have been previously reported by Vithani et al. [27]. Drug-loaded SMEDDS were prepared following a previously reported process and formulation [27] (Table 4 and Table 5), where FEN was used as the model drug. In this study, FEN was used for the single compartment PLHs in SMEDDS A, while clofazimine (CLO), lumefantrine (LUM) and halofantrine (HAL) were used for the multicompartment systems, where each of the six drug/SMEDDS combinations were individually prepared and loaded separately into a ‘wedge’ in the dose form. It is important to realise that 3DP systems with FDM and SSE heads in the one unit have been reported [37], and that the manual preparation method is sufficient to test the concept of controlling dispersion and release from varying geometries and from independent compartments in the multicompartment system [36].

Batches of drug-loaded SMEDDS for use in single compartment studies were prepared in quantities ranging from 10 to 15 g (Table 4.), and individual drug batches for multicompartment studies were prepared in quantities of 3 g (Table 5).

The process for preparing a batch of drug-loaded SMEDDS is illustrated in Figure 2. Gelucire® 44/14 (GEL 44) was weighed into a glass beaker which was then placed onto a heated hotplate stirrer set to 80 °C (melting point of FEN) and stirred using a magnetic stirrer bar. Gelucire® 48/16 was deposited into the molten GEL 44. Then, Kolliphor P 188 was deposited into the mixture in a similar manner. Lastly, the drug component was deposited into the mixture and stirred for 30 min to ensure homogeneity. The mixture was then transferred to a 50 mL polypropylene tube and immediately sonicated for 2 min in a sonication bath. This was done to remove any air bubbles from the molten SMEDDS.

#### 2.2.2. Filling 3DP Scaffolds with Molten Lipid Formulation

Five different tablet formats possessing a range of surface area-to-volume ratios (SA:V) were constructed for the study of single-compartment systems (Table 6). Each tablet format refers to a different scaffold (or scaffolds), printed with PLA (Figure 3A) or PVOH (Figure 3B), that has been filled with a molten SMEDDS formulation. SA:V values were calculated on the assumption that the theoretical cavity volume (392.6 µL) was occupied by the SMEDDS. ‘Semi-open’ and ‘Closed’ tablet formats required scaffold covers (Table 2A,B, respectively) in addition to a scaffold container (Table 1C), which were manually attached after the SMEDDS had solidified.

A target mass of 400 mg molten drug-loaded SMEDDS was pipetted into a polymer scaffold, whilst the scaffold remained on the weighing plate. For the filling of ‘no-scaffold’ type systems, a silicon mould made from the scaffold cavity was used instead (See Appendix A). For the filling of ‘dual-face’ scaffolds, the scaffold was attached to a cold smooth surface before filling (See Appendix A).

#### 2.2.3. Filling Multicompartment Scaffolds with Molten Lipid Formulation

Multicompartment tablet systems consisted of CLO, HAL and LUM incorporated into compositions 1 and 2 (Table 5) to uniquely fill each ‘wedge’ with a target of 66 mg of molten drug-loaded SMEDDS, as indicated in Figure 4 below. The filled mass for each wedge is provided in the Appendix A.

#### 2.2.4. Preparation of Mixed Drug and Formulation Pellets

Pellets without a scaffold were also prepared from a mixture of all the drug:lipid compositions used in the multicompartment systems as control systems (Figure 5). The molten mixture was deposited into a silicon mould, solidified and the pellet pressed out of the mould to generate solid mixed drug/formulation pellets without a scaffold.

### 2.3. Dispersion Studies in Simulated Gastric to Intestinal Media

In vitro dispersion studies in sequential gastric and intestinal media were conducted as illustrated in Figure 1 (in vitro dispersion testing). Titration vessels (Metrohm titration vessel with thermostat jacket 20–90 mL) were connected to a heated water pump, set to 37 °C (Ratek immersion heater and insulation tank), using rubber tubing. The jacketed vessels were placed on a multipoint magnetic stirrer (Daihan Scientific, Wonju, Korea) set to 300 rpm with one magnetic stirrer bar placed in each vessel (L. 30 mm, bar diam. 8 mm).

A 3D-printed PLA mount (see Appendix A) was used to attach to and submerge the scaffolds containing the formulations. The weight of the scaffolds, formulations and drug mass for each system are detailed in the Appendix A. For the first 30 min, the formulation was submerged in simulated gastric fluid (gastric fluid herein) followed by a 3-min transition window, then 30 min of fasted state simulated intestinal fluid (intestinal fluid herein). Gastric fluid (0.1 M HCl, pH 1.2) was deposited into the titration vessel 5 min prior to dispersion, to allow for heat transfer. Aliquots (1 mL) were taken at timepoints; 1, 3, 5, 10, 15, 20, 30, 34, 26, 38, 43, 53, 63 min after immersion (t_1_–t_14_). Aliquots were immediately transferred to 1.5 mL Eppendorf tubes and stored for analysis. At the end of the gastric phase, 3.2 mL of 0.95 M NaOH was added to the vessel to neutralise the gastric fluid, followed by 6.4 mL of ‘Concentrated FaSSIF solution’ (Table 7), and finally 6.4 mL of deionised water to establish the simulated intestinal condition. 

The concentrated FaSSIF solution was prepared using amounts of each component at 4.5 times the amount needed for physiological relevance to allow for dilution.

### 2.4. Dispersion Kinetics of SMEDDS Studied Using Nephelometry

Samples retrieved at the timepoints indicated in Section 3.3 were vortexed for ~5 s before aliquoting 350 µL of the dispersion into the well of a 96-well UV-transparent microplate. Turbidity was determined using a Nephelostar Plus nephelometer (BMG Labtech) set to 37 °C, with a laser beam focus of 2.5 mm and laser intensity of 80%. A settling and interval time of 1 s each was allowed between measurements. Furthermore, the sample plate was agitated using the double orbital shaking option for 1 s at 400 rpm, prior to each measurement. The total formulation dispersed (%) in gastric or intestinal fluid was determined by extrapolation from a dispersed FEN:SMEDDS calibration curve.

### 2.5. Quantification of Drug in Dispersion Media—UPLC

Drug concentrations in dispersion media were quantified by UPLC for the four different drugs used in this study. The details for methods and equipment are detailed in the Appendix A.

### 2.6. Solid State Characterisation - X-ray Diffractometer (XRD)

A Shimadzu X-ray Diffractometer (XRD-7000L) was used to measure the changes in diffracted X-ray intensities of solid samples, at a rotation angle range of 10–90°. Samples were deposited and smeared into the sample tray such that the sample was flush with the sample tray height. The X-ray diffraction system was set to a horizontal gonio type and data were processed using XRD-6100/7000 software (XRD-series system, version 7.0.1.1, 2014, Shimadzu, Rydalmere, Australia).

### 2.7. X-ray CT Imaging Analysis

X-ray computed tomography (CT) was conducted using a Zeiss Xradia 520 Versa and image files were post-processed using Dragonfly Pro software (version 2021.1.0.997, 2021, Object Research Systems (ORS) Inc., Montreal, QC, Canada) [38].

### 2.8. Cryo-TEM Imaging Analysis

Cryogenic transmission electron microscopy (Cryo-TEM) was used to analyse ~400 mg of a blank SMEDDS dispersed in 20 mL of deionised water. A total of 3.5 µL of the dispersion was spotted onto a TEM grid (EM Lacey Formvar Cu 300 mesh, 50 µm) followed by vitrification on a FEI Vitrobot Mark IV (blot force = -4 and blot time = 3). Images were collected on a 120 keV FEI Tecnai Spirit G2 TEM at a low-dose mode. Dispersion samples for cryo-TEM analysis were prepared and imaged on the same day.

## 3. Results

### 3.1. Characterisation of SMEDDS

The X-ray diffraction patterns in the Appendix A illustrate the crystalline nature of the lipids in the solid SMEDDS formulation, as the same peaks are apparent in the diffractograms for both the drug-free and drug-loaded SMEDDS. There were very minor diffraction peaks apparent for the SMEDDS where drug has been previously dissolved upon heating (‘Mixed’) at the positions of the major peaks in the corresponding reference crystalline drug (top profile). A physical mixture of the blank SMEDDS with the reference material showed lipid peaks at similar height to the ‘dissolved’ SMEDDS, and also had more prominent peaks corresponding to the drug, indicating that at least a proportion of the drug was present in a crystalline form. The representative cryo-TEM image shown in Appendix A displays the colloidal structures present in the media after dispersion of a blank SMEDDS formulation. The blank SMEDDS dispersed in water to form a colloidal dispersion with droplet size of ~ 5–10 nm.

### 3.2. Characterisation of Filling of the Single Compartment Scaffolds Using X-ray CT

X-ray CT images were processed to isolate a region of interest (ROI) that defines the apparent void area between the lipid filling and PLA wall of a FEN-loaded PLA PLH tablet (Figure 6). Analysis of the PLA scaffolding and SMEDDS formulation alone was also conducted (Appendix A).

Figure 6C displays the increase in void area at the PLA–SMEDDS interface, over a ~5 mm height range between slices at 219 and 723 µm. This slice range aims to capture the void area from the floor of the inner compartment to the underside of its lid. Moreover, the interfacial gap at the interface between the lipid filling and PLA wall increases with tablet height, demonstrating the immiscibility of the two materials.

### 3.3. Dispersion of Drug-Loaded SMEDDS Formulations from Single Compartment Scaffolds Using Turbidity

The dispersion kinetics of SMEDDS formulations contained within the PLA and PVOH scaffolds when dispersed in gastric to intestinal in vitro conditions are illustrated in Figure 7, Figure 8, Figure 9, Figure 10. The percentages reported are calculated relative to the turbidity achieved using a completely dispersed non-scaffolded SMEDDS of the same mass.

Non-dissolving PLA scaffolds: The SMEDDS without a scaffold, with the highest SA:V of 0.8, reached a peak turbidity of 98% dispersed within 10 min (essentially complete). The semi-open tablets reached a level of dispersion approximately 78% of the fully dispersed control SMEDDS within 30 min. Closed type PLA-based tablets displayed essentially no release and appeared to remain closed throughout the dispersion. During the intestinal phase of the dispersion, the no-scaffold and single-face types plateaued at ~100% dispersed, while the dual-face types plateaued at ~110%. The semi-open type tablets plateaued at ~90% (albeit with large error bars due to variation apparent for individual units). The order of dispersion rate ranked as expected as: no-scaffold > dual-face > single-face > semi-open, closed did not disperse.

Dissolvable PVOH scaffolds: The dispersion of SMEDDS from dissolvable PVOH scaffold tablets is shown in Figure 8. The no-scaffold tablet data from Figure 7 are identical to Figure 8. A maximum of ~77% was released within 30 min for the semi-open type scaffolds. The closed-type PVOH-based tablets displayed significant release between 20 and 30 min (because of the dissolution of the polymer compared to PLA), at which point the lids had visually come open during dispersion. During the intestinal half of the dispersion, all tablet types plateaued at ~100% dispersion, including the closed type. The rank order of dispersion rate from highest to lowest for the PVOH scaffolds therefore was the same as the PLA tablets; however, the erosion of the closed PVOH system enabled release that did not occur for the non-eroding PLA (at least on the current timescale).

### 3.4. Release of Drug (FEN) from Single Compartment Tablets

The effect of tablet format on the release kinetics of FEN from the PLA-based scaffolds, determined using HPLC, is demonstrated in Figure 8. FEN was released almost completely from the no-scaffold-type tablets (SA:V = 0.8) within 10 min. The semi-open type formulations released 30% of the FEN within 30 min, while practically no FEN was released from the closed-type PLA-based tablets which as mentioned in the turbidity studies remained closed throughout the entire dispersion. During the intestinal phase of the dispersion, the drug release from the no-scaffold-type tablets plateaued at ~105% while the dual-face and single-face scaffolds increased to between 85% and 105% of FEN. The semi-open-type tablets demonstrated variable but continual release of FEN throughout gastric and intestinal phases. The order of release of FEN from the scaffolds logically followed the order of dispersion of the SMEDDS.

Similar behaviour was again observed for drug release when PVOH was used for the scaffold where release from the fully closed scaffold took around 20 min to commence but was complete in the intestinal phase (Figure 10).

### 3.5. Release of Drugs from Multi-Compartment Tablets

The release kinetics of model drugs HAL, LUM and CLO released from multicompartment tablets (Figure 4) and mixed drug pellet tablets (Figure 5, where all compartments of the multicompartment tablet have effectively been mixed into one) are displayed in Figure 11. Drug release from the mixed drug pellet was faster than release from the multicompartment system. The drug release from multicompartment tablets was approximately linear and directly proportional to time spent in dispersion media. The release of LUM from multicompartment tablets was slower than the other drugs, while all drugs from the mixed drug pellets appear to release at a similar rate and plateau after 20 min. By 20 min, the release of HAL, CLO and LUM reached 90%, 80% and 75% of total mass, respectively. In contrast, the release of HAL, CLO and LUM from the multicompartment systems reached 60%, 75% and 40% of total mass, respectively, at 30 min. Complete dispersion of the pellet could not be determined visually due to the high opacity of the dispersion media.

## 4. Discussion

The developments in three-dimensional (3DP) printing within the realm of pharmaceutics to generate orally administered dosage forms has, to date, been dominated by polymer-based formulations. Lipid-based systems have been proven to boost the bioavailability of lipophilic drugs, yet research into the use of lipid-based formulations in 3D printing is in its infancy. Vithani et al. developed a proof of concept for a semi-solid lipid-based formulation with self-micro emulsifying properties that could be 3D-printed via soft material extrusion [27]. However, the printing fidelity was not sufficiently high to translate to viable printed medicines and this system required further optimisation to better control its dispersion kinetics under gastric and intestinal conditions. 

To address this shortcoming, the potential of 3D-printed polymer scaffolds containing the lipid-based SMEDDS formulation to control print fidelity and exposed surface area was explored. Different polymer scaffolds (Table 1, Table 2, Table 3) were printed using a fused deposition modelling 3D printer with PVOH and PLA as separate extrudates, using a process that resembles previous reports of building PVOH capsules from 3DP core shell design [39]. This approach allowed much higher fidelity dosage units without the ‘slumping’ apparent when printing SMEDDS systems without a scaffold [27].

The single-compartment scaffolds were filled with FEN-loaded SMEDDS to generate a range of 3D-printed PLH-based tablets with increasing SA:V ratios by modifying the exposed surface area (Figure 3 and Figure 4). For this study, the SMEDDS was manually filled into the scaffolds, which while not practical in a manufacturing context enabled the proof-of-concept study and determination of the relationships between exposed surface area, volume, and polymer dissolution to be investigated. There are reports in the literature of combined FDM and semi-solid extrusion printers with the two different print heads in the printer [2,19]. Hence, it would be possible to manufacture the PLH tablets in one operation to achieve this capability.

The dispersion rate of the solid SMEDDS into gastric media was directly proportional to the SA:V ratio when considering the amount of formulation dispersed at 20 min for each tablet (Figure 12A). Interestingly, there was no difference between the dispersion rates for the PLA and PVOH scaffolds, indicating that dissolution of the PVOH scaffold occurred more slowly than dispersion of the formulation. Note that the closed PVOH scaffold had not released formulation at the 20 min mark but was clearly differentiated from the PLA scaffold at later timepoints (Figure 10).

Fenofibrate is a poorly water-soluble drug, hence release of drug into the dissolution media would be expected to follow the kinetics of dispersion. The relationship between SA:V and drug release kinetics from solid dosage forms has been studied for 3D-printed polymer-based tablets [17,40]. In this study, a general increase in drug release as the amount of formulation dispersed increased was observed at 20 min (Figure 12B). However, the amount of drug released was slightly lower than what would be expected for a 1:1 correlation with formulation dispersion.

Scalability of printed dose forms to suit the patient is a major advantage of 3D-printed dose forms. Of course, the large scaffolds printed here would be too large for a patient—this was merely an arbitrary size selected for easy manual handling when assembling dispersion and dissolution experiments.

The additive manufacturing approach offers an interesting opportunity to provide tailored release behaviour of multiple drugs from the one dose unit. The concept of a polypill was first coined in 2003 by Wald et al. as a combined therapy daily pill to treat cardiovascular disease [29] and has been used to describe orally administered dose forms containing multiple active ingredients. The use of polypill approaches has been shown to increase adherence rates, improve cost effectiveness of therapies and increase the safety of medicines [41]. There have been reports of 3DP polypills with defined release profiles [15,32], and 3DP dual-compartment lipid-based tablets with sequential release kinetics of two drugs [36]. However, to our knowledge there has not been any reports of 3DP solid lipid-based polypills. Pereira et al. published a novel multicompartment 3D-printed system that shows promise for testing and delivery of fillable formulations with defined release rates [34].

Conceptually, printing one drug as a single compartment is as challenging as printing multiple drugs when they are printed as multiple isolated compartments. It is believed, with the advent of 3D-printed products, these challenges are perhaps less of an issue than might initially be considered compared to combination dose forms where the drugs are mixed together.

The release behaviour from the polypills was different to the single-compartment systems—release of drug was approximately linear with time. This is suggestive that the formulation is gradually dispersed into the media based on the available surface area of the formulation, which is essentially dictated by the wedge shape exposed to the dispersion media. Steady erosion of the formulation from within the wedge-shaped holes then is expected to be faster than the erosion of the PVOH scaffold. This was confirmed by retrieval of multicompartment systems at different timepoints as seen in the photos in Figure 13. SMEDDS A (composed of Gelucire 48/16 combined with other surfactants) and B (only composed of Gelucire 48/16) were used in the study, but it was not possible to discriminate release from either SMEDDS because total amount of each drug simultaneously from the two formulations was quantified together. Thus, it is possible that the ‘linear’ profiles result from a fast and slow contribution from the different SMEDDS, however the erosion evident in the photos in Figure 13 would suggest that SMEDDS A and B dispersed at a similar rate.

X-ray CT analysis of the void area, or the gap, between the PLA scaffold and SMEDDS formulation increased with height through the tablet. This suggests that these two materials are not miscible. In addition, the increase in void area with height may be attributed to shrinking of the SMEDDS as it cools within the scaffold. Furthermore, the porosity of the PLA scaffolding is not uniform, and a particularly large difference in porosity may be observed between the foot and walls of the scaffolding (Figure 6A). It is unclear whether porosity has influenced the dispersion and subsequent release of the SMEDDS from PLA formulations; however, it is believed to have a negligible effect considering the rapid dispersion time and hydrophobicity of PLA. This demonstrates the need for more research into miscible 3DP PLA-SMEDDS core shell designs, and to optimising the printing resolution to eliminate porosity, that may be achievable with PLH-type filament and continuous printing methods [42].

The model drugs for the multicompartment combination study were selected based on lipophilicity, not direct application of the combination. Clofazimine, halofantrine and lumefantrine each possess high logP values of 5.2, 7.6, 8.6 and 8.6, respectively (Pubchem). In a clinical setting, all three are anti-infective drugs, with clofazimine being used for leprosy [43], and lumefantrine and halofantrine to treat malaria [44,45] Multidrug combinations for anti-infective treatments are common and the demonstration of incorporating relevant combinations of drugs in the one dose form, potentially in different formulations to control dispersion and release opens new opportunities for convenient treatment of such conditions in clinically challenging settings.

The drugs appear to be partly crystalline in the solid SMEDDS at room temperature, and the X-ray diffraction data at least for fenofibrate support this assertion. Our experience with these compounds is that digestion of lipid formulations favours drug solubilisation [46,47]. Here, the focus was on control of dispersion of the formulation and drug and less on its solid state upon dispersion, so the effects of digestion were not studied specifically in this study. There is, however, sufficient literature evidence that digestion will stimulate solubilisation of already crystalline drug, not precipitation as might be expected for a liquid SMEDDS formulation where the drug is fully dissolved prior to dispersion [48].

The translation of this concept into a product will require the development of processes at sufficient speed so that it becomes viable to print such systems. It should be noted that 3D printers have been reported that have an FDM and semisolid extrusion nozzle in the one printer [49], meaning it is technically possible to prepare these types of combination dose forms without manually filling the scaffold. Furthermore, it is difficult to predict the output speed of a dosage form based on the complexity of its design alone, as the speed and resolution of deposited materials is highly dependent on the material itself and the printing method used. Therefore, the current application of such a formulation is more likely to serve a niche requirement for bespoke dose forms generated in a hospital or doctor’s surgery, rather than in large scale additive manufacturing. Nevertheless, it is possible that the continual development of 3D printing technology will realise larger scale additive manufacturing of bespoke drug forms over time.

## 5. Conclusions

This study demonstrates that the gastric-to-intestinal dispersion rate of solid lipid-based formulations may be controlled using 3D-printed scaffolding. Furthermore, it demonstrates that an entirely 3D-printed biodegradable PLH tablet, using poly-vinyl alcohol and SMEDDS, may be modified to deliver bespoke and combined therapies in a single multi-compartment tablet. The versatility of these 3D-printed PLH tablets has the potential to circumvent challenges associated with the adherence and safety of polypharmacy, together with the 3D printing of lipophilic drugs.

## Figures and Tables

**Figure 1 pharmaceutics-13-02107-f001:**
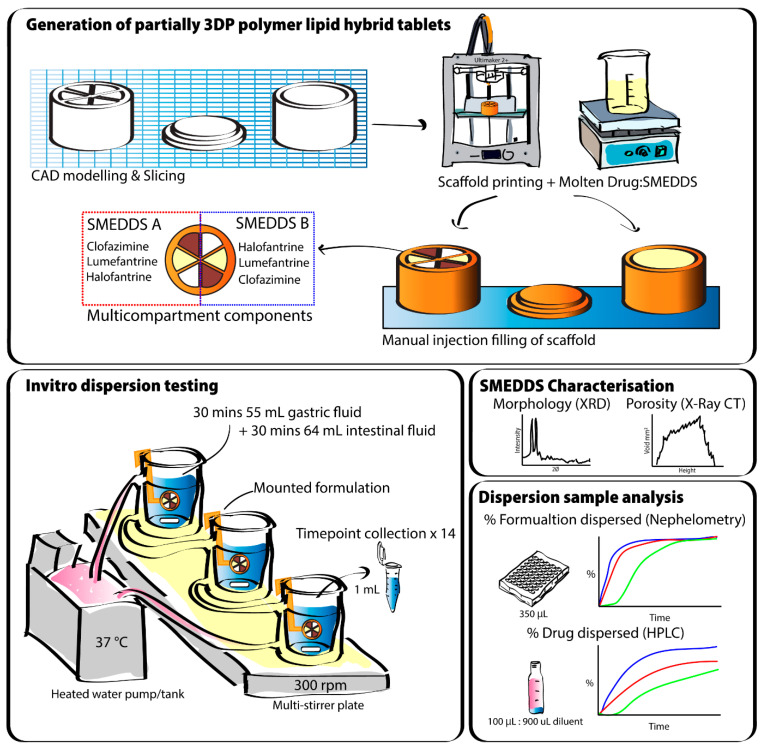
Schematic displaying the general approach to prepare and investigate the polymer–lipid hybrid tablets. In the top panel, the printing of single- and multicompartment scaffolds is illustrated. The bottom panels illustrate the dispersion and release methodologies and analytical techniques.

**Figure 2 pharmaceutics-13-02107-f002:**
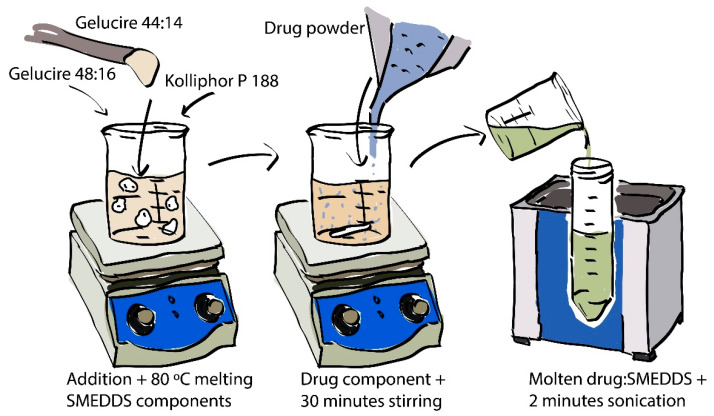
Illustrated process for the preparation of drug-loaded SMEDDS.

**Figure 3 pharmaceutics-13-02107-f003:**
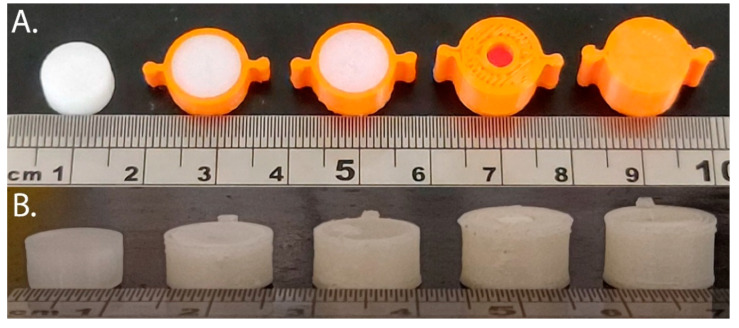
Tablet formats with decreasing exposed surface-area to-volume (SA:V) ratios from left to right generated from PLA **(A)** and PVOH (**B**). Tablet formats from left to right: no-scaffold, dual-face, single-face, semi-open, closed.

**Figure 4 pharmaceutics-13-02107-f004:**
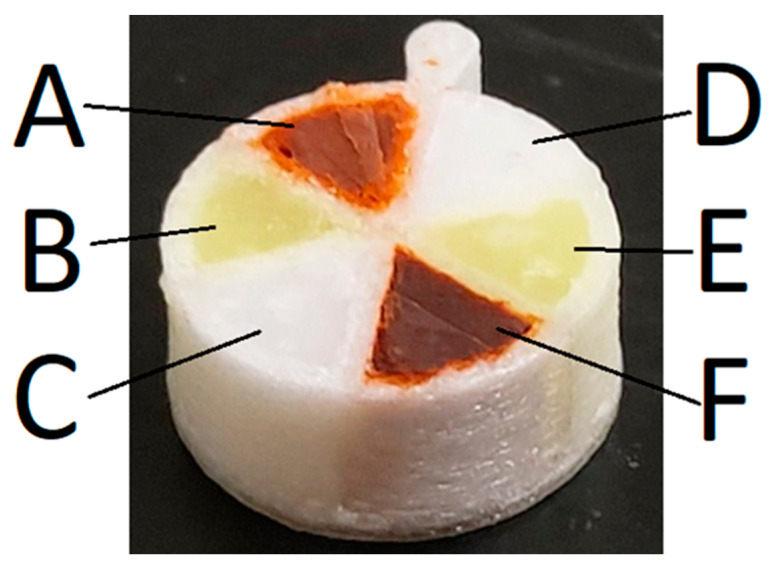
3D-printed multicompartment PLH constructed from PVOH containing drug + formulation (**A**) 7% CLO:SMEDDS, (**B**) 7% LUM:SMEDDS, (**C**) 3.5% HAL:SMEDDS, (**D**) 7% CLO:Gelucire® 48/16, (**E**) 7% LUM:Gelucire® 48/16, and (**F**) 3.5% HAL:Gelucire® 48/16.

**Figure 5 pharmaceutics-13-02107-f005:**
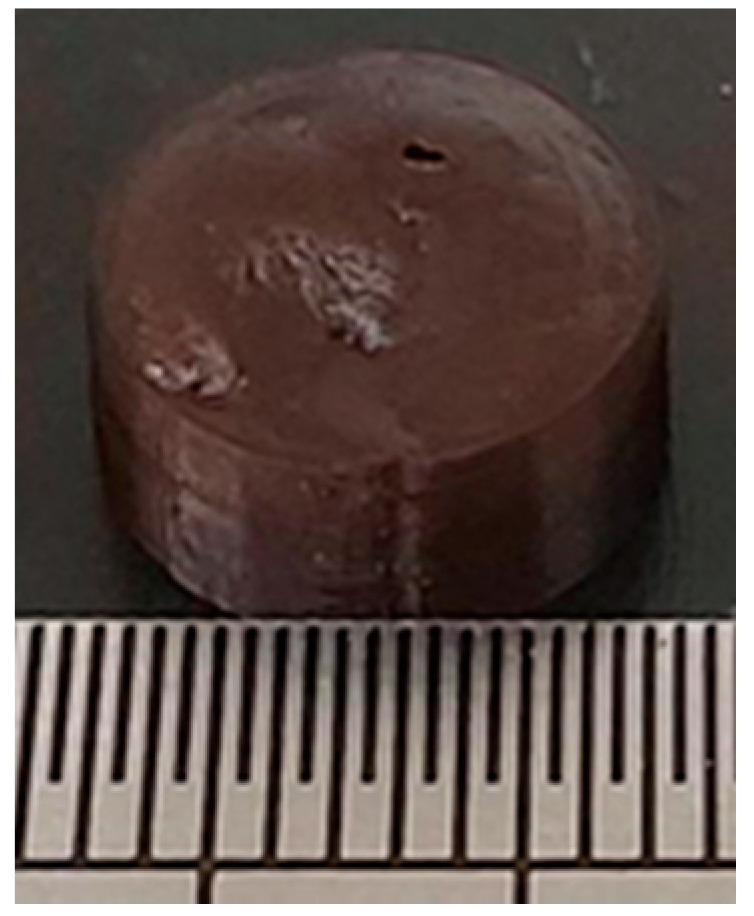
‘Control’ pellet containing drug and formulation combinations used in multicompartment PLHs. Composed of 82.5% 1:1 SMEDDS + Gelucire^®^ 48/16, with 7% clofazimine, 7% lumefantrine and 3.5% halofantrine. (Scale is 1 mm per fine graduation).

**Figure 6 pharmaceutics-13-02107-f006:**
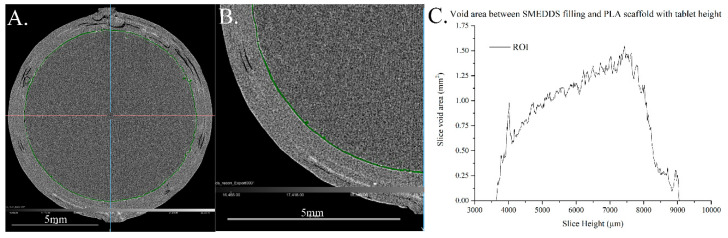
Horizontal cross-section images acquired via X-ray CT imaging of a FEN-loaded PLA PLH, with void area ROI highlighted in green. (**A**) Full scale cross-section image at slice 505, (**B**) single quadrant cross-section image at slice 505. (**C**) Area analysis (mm^2^) of ROI with increasing slice height (µm) of a FEN-loaded PLA-based PLH.

**Figure 7 pharmaceutics-13-02107-f007:**
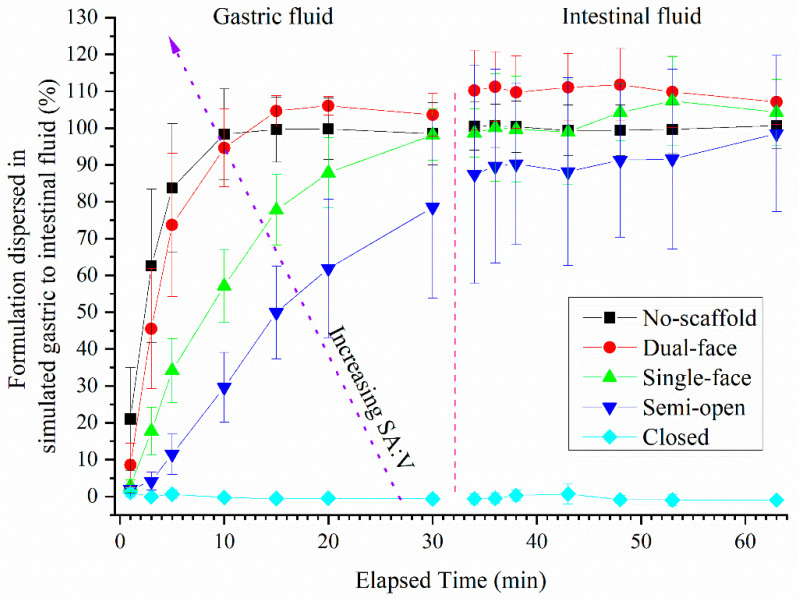
Turbidity of dispersion media resulting from dispersion of fenofibrate-loaded SMEDDS from different single-compartment PLA-based scaffolds (error bars—AVG ± SD, whereby *n* = 7 for no-scaffold, *n* = 4 for dual-face and semi-open, *n* = 3 single-face and closed-type tablets). The dashed red line indicates the transition from gastric to intestinal conditions. The 100% level was determined by full dispersion of a blank SMEDDS. Pellet without any polymer scaffold.

**Figure 8 pharmaceutics-13-02107-f008:**
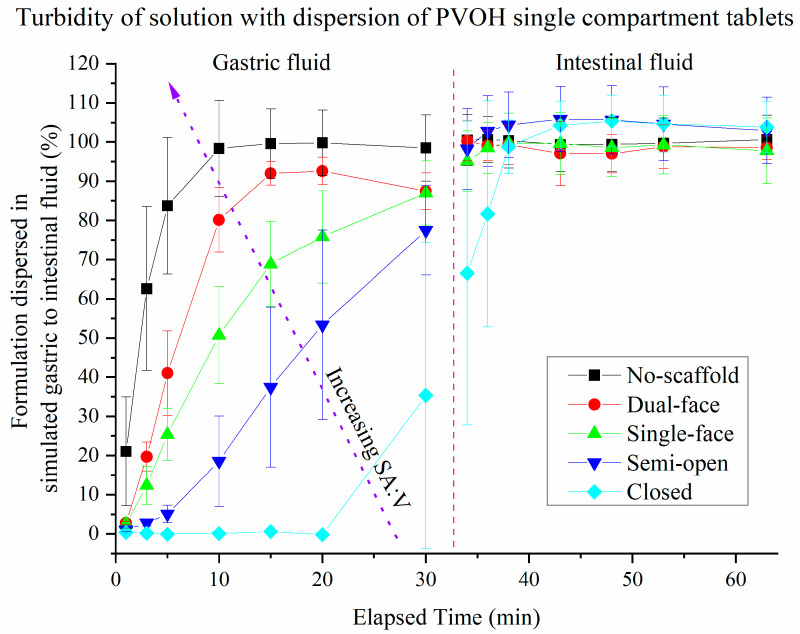
Turbidity of dispersion media resulting from dispersion of fenofibrate-loaded SMEDDS from different single-compartment PLA-based scaffolds (error bars—AVG ± SD, whereby *n* = 7 for no-scaffold, *n* = 4 for dual-face, semi-open and single-face, and *n* = 3 for closed type tablets). The dashed red line indicates the transition from gastric to intestinal conditions. The 100% level was determined by full dispersion of a blank SMEDDS. Pellet without any polymer scaffold.

**Figure 9 pharmaceutics-13-02107-f009:**
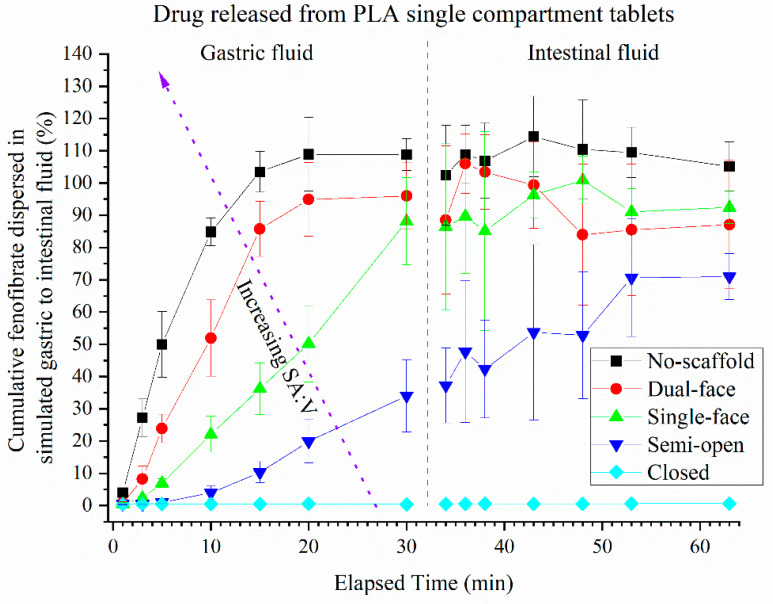
Effect of exposed surface area on release of FEN from different single-compartment PLA-based scaffolds containing fenofibrate-loaded SMEDDS measured by HPLC (error bars = AVG ± SD, *n* = 3, with exception of no-scaffold series whereby *n*=6). The dashed red line indicates the dispersant transition period.

**Figure 10 pharmaceutics-13-02107-f010:**
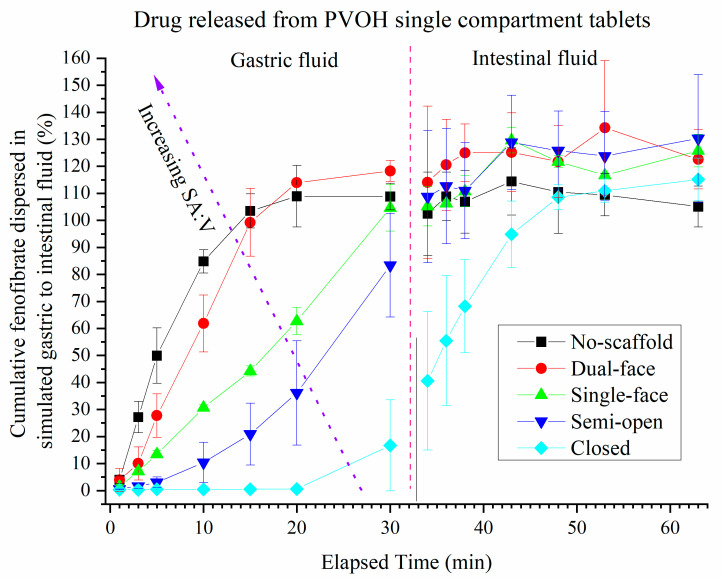
Drug release from different single-compartment PVOH-based scaffolds containing fenofibrate-loaded SMEDDS measured by HPLC. (Error = AVG ± STD, *n* = 4 for closed and single face series, *n* = 3 for semi-open and dual-face series, *n* = 6 for no-scaffold series). The dashed red line indicates the dispersant transition period.

**Figure 11 pharmaceutics-13-02107-f011:**
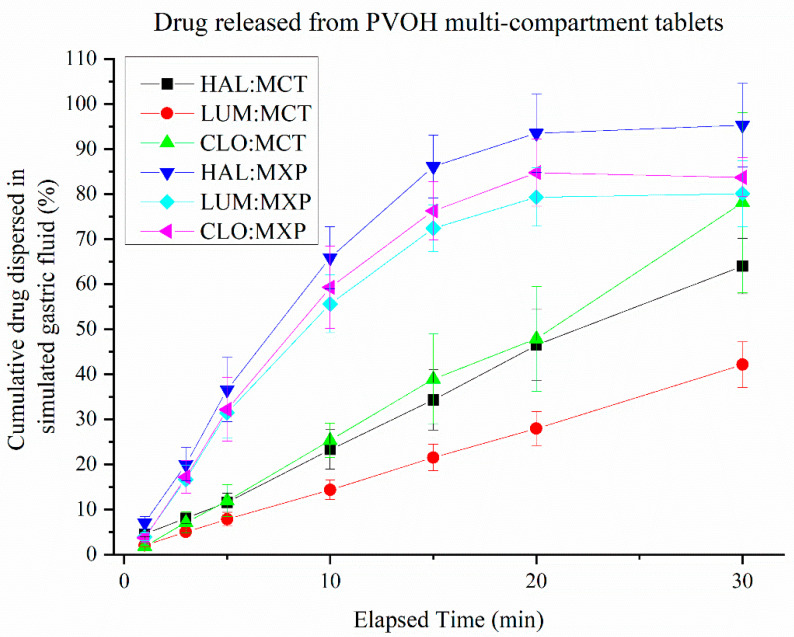
Drug release from multicompartment PVOH-based scaffold tablets (MCT), and mixed drug pellets (MXP), containing HAL-, LUM- and CLO-loaded SMEDDS (error bars = AVG ± STD, *n* = 4).

**Figure 12 pharmaceutics-13-02107-f012:**
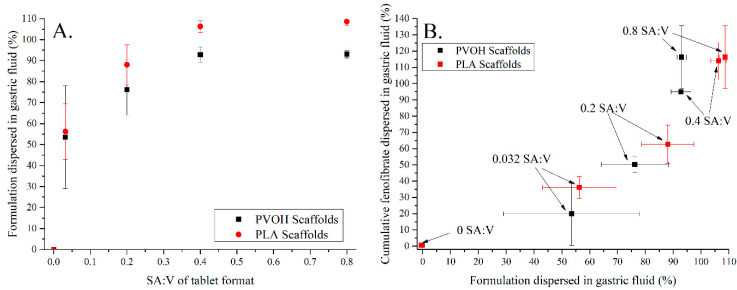
(**A**) Relationship between dispersion of the formulation and surface area-to-volume ratio (SA:V) and(**B**) the dependence of drug release on dispersion of the formulation across the range of SA:V studied. Black symbols indicate PVOH-based scaffolds; red symbols indicate PLA-based tablets after 20 min of dissolution time in simulated gastric media.

**Figure 13 pharmaceutics-13-02107-f013:**
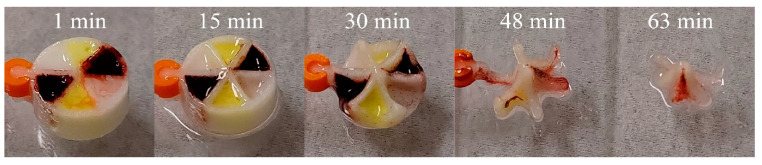
Photo series demonstrating the erosion of PVOH multicompartment tablets when retrieved form the dispersion medium at different time points (see Appendix A for full photoset).

**Table 1 pharmaceutics-13-02107-t001:** Geometry of virtual PLA and PVOH-based scaffold containers used to print physical scaffold containers. Measurements for PVOH–based scaffolds have been underlined. A, B, C refer to different exposed surface area scaffolds with equal internal volume.

Scheme	A 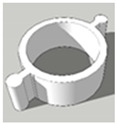	B 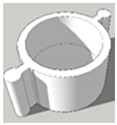	C 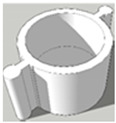
**Foot diameter (mm)**	12, 11.3	12, 11.3	12, 11.3
**Height (mm)**	5, 5	7, 5.6	8, 6.2
**Cavity depth (mm)**	5, 5	5, 5	6, 5.6
**Cavity diameter (mm)**	10, 10	10, 10	10, 10
**Wall thickness (mm)**	1, 0.65	1, 0.65	1, 0.65
**Printing time (min)**	2.5	5	5

**Table 2 pharmaceutics-13-02107-t002:** Geometry of virtual PLA and PVOH-based scaffold covers used to print physical scaffold covers. Measurements for PVOH–based scaffolds have been underlined. A and B refers to the two different (open or closed) lids.

Scaffold Cover	A 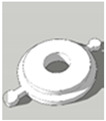	B 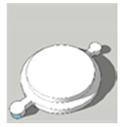
**Foot diameter (mm)**	12, 11.3	12, 11.3
**Overall height (mm)**	2, 1.2	2, 1.2
**Lip height (mm)**	1, 0.6	1, 0.6
**Lip thickness (mm)**	1, 0.65	1, 0.65
**Cavity diameter (mm)**	4, 4	NA, NA
**Printing time (min)**	2.5	2.5

**Table 3 pharmaceutics-13-02107-t003:** Geometry of virtual multicompartment scaffolds used to print physical multicompartment scaffolds from PVOH filament.

Multicompartment Scaffold (Constructed from PVOH)	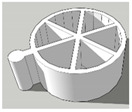
**Foot diameter (mm)**	12.4
**Wedge depth (mm)**	5
**Height (mm)**	5.6
**Wall thickness (mm)**	0.6
**Printing time (min)**	6.25

**Table 4 pharmaceutics-13-02107-t004:** Compositions of the FEN-loaded SMEDDS used for the single-compartment systems.

Single Compartment Systems	
SMEDDS A	Component (% *w*/*w*)
Drug	FEN
Drug content	7.0
Gelucire^®^ 44/14	46.5
Gelucire^®^ 48/16	23.3
Kolliphor P 188	23.2

**Table 5 pharmaceutics-13-02107-t005:** Compositions of the two different drug-loaded SMEDDS formulations used for the multicompartment systems.

Drug	SMEDDS A			SMEDDS B		
	CLO	LUM	HAL	CLO	LUM	HAL
Drug content	7.0	7.0	3.5	7.0	7.0	3.5
Gelucire® 44/14	46.5	46.5	48.3			
Gelucire® 48/16	23.3	23.3	24.1	93.0	93.0	96.5
Kolliphor P 188	23.2	23.2	24.1			

**Table 6 pharmaceutics-13-02107-t006:** List of decreasing SA:V ratios for respective tablet formats, in mm^2^ × µL^−1^.

Tablet Type	No-Scaffold	Dual-Face	Single-Face	Semi-Open	Closed
**SA:V ratio (mm^2^ × µL^−1^)**	4:5 (0.800)	2:5 (0.400)	1:5 (0.200)	4:125 (0.032)	0:1 (0.000)

**Table 7 pharmaceutics-13-02107-t007:** Components required for 100 mL of concentrated FaSSIF solution.

Component	Mass Required (g)
NaH_2_PO_4_	3.438
NaOH pellets	0.420
NaCl	6.186
FaSSIF powder	2.240

## Data Availability

Not applicable.

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
