# Peer review of "A 3D-Printed Polymer–Lipid-Hybrid Tablet towards the Development of Bespoke SMEDDS Formulations"

_pharmaceutics, 2021, doi:10.3390/pharmaceutics13122107_

Round 1

Reviewer 1 Report

Title- A 3D Printed Polymer-Lipid-Hybrid tablet towards the development of bespoke SMEDDS formulations

In this work the author formulated and evaluated 3D printed solid lipid formulations by providing a 3D printed dissolvable polymer scaffold to compartmentalize solid lipid formulations within a single dosage form. This is a good work and well written manuscript. Manuscript provides information as well as discussion in good detail. However, there are some minor comments which need to be addressed by the author. My recommendation is “Minor Revision” Refer to my comments below.  

Comments-

  1. Did author consider the effect using different polymers? If so, which were considered. And is the polymer solubility in physiological pH was the only criteria considered during selection of polymers?
  2. Formulation application-
    1. Are these formulations meant for hospital pharmacy and dispensing pharmacy? Or these were made considering its scalability at larger scale?
    2. If these were considered for larger scale manufacturing then author should discuss the limitations because of followings,
    3. complex design will lower the output
    4. cost associated to SMEDDS development
    5. very hard to accurately dose and maintain constant content uniformity for multiple drug loading
    6. stability profile and sensitivity for individual loaded drug’s will be different
    7. issues related to determine %API content in combinational drugs considering these designed formulations are meant for >3 drugs
  3. Patient compliance will be the issue because of big size tablets, this should be discussed
  4. Preparation of mixed drug and formulation pellets- what is author’s opinion on stability because this kind of system will be stable only
    1. if the components are not interacting with each other to cause chemical degradation?
    2. In SMEDDS system all the used drugs have good solubility
  5. Characterization of SMEDDS- author stated that proportion of the drug was present in a crystalline form. This could be potential issue related to the stability of SMEDDS because small amount of crystalline compound may behave as a seed and based on different physiological environment it may lead to recrystallization of significant drug used. Suggested to provide brief information about the drug’s saturation solubility in used SMEDDS excipients

Reviewer 2 Report

The manuscript entitled "A 3D Printed Polymer-Lipid-Hybrid tablet towards the development of bespoke SMEDDS formulations" provides sound discussion with a reasonable set of experimental designs. Indeed, the manuscript needs to be revised considering the following suggestions.

  1. It is suggested to include the different drugs incorporated in the developed formulation should be mentioned in the abstract. The present form of abstract provides a generalized overview, I will suggest to revised and make a little more specific incorporating the absolute results.
  2.   It is advised to incorporate the clinical basis of selection clofazimine, lumefantrine, halofantrine for combination pharmaceutical product development in the introduction section.  Also, incorporate the physicochemical properties of these drugs which favors its design as SMEDDS formulation in the introduction section.
  3.  The authors have not mentioned the droplet size distribution and zeta potential profile of the developed SMEDDS system as combination products. It is suggested to include these results. 
  4. It will be interesting for the reader to show that developed combination formulation does not cause drug precipitation in gastrointestinal fluid.  In my opinion, such supporting results highlight the stability of the drug in GI fluid and not cause drug precipitation should be incorporated. 
  5. Authors have to include the future prospects of such kind of 3D printed formulation system and suggest some directions which will be helpful to reach clinical settings for personalized medications. Also suggests an alternative to manual feeding of SMEDDS system in FDM printed scaffolds. 

Round 2

Reviewer 2 Report

The revised manuscript should be accepted for publication in present form.